# Serological Evidence of Hepatitis E Virus Infection in Semi-Domesticated Eurasian Tundra Reindeer (*Rangifer tarandus tarandus*) in Norway

**DOI:** 10.3390/pathogens10121542

**Published:** 2021-11-25

**Authors:** Christine Hanssen Rinaldo, Ingebjørg Helena Nymo, Javier Sánchez Romano, Eva Marie Breines, Francisco Javier Ancin Murguzur, Morten Tryland

**Affiliations:** 1Department of Microbiology and Infection Control, University Hospital of North Norway, N-9038 Tromsø, Norway; 2Metabolic and Renal Research Group, Department of Clinical Medicine, UiT The Arctic University of Norway, N-9037 Tromsø, Norway; 3The Norwegian Veterinary Institute, N-9016 Tromsø, Norway; ingebjorg.nymo@vetinst.no; 4Department of Arctic and Marine Biology, UiT The Arctic University of Norway, N-9037 Tromsø, Norway; javier.s.romano@uit.no (J.S.R.); eva.m.breines@uit.no (E.M.B.); francisco.j.murguzur@uit.no (F.J.A.M.); 5Department of Medical Biology, UiT The Arctic University of Norway, N-9037 Tromsø, Norway; 6Department of Forestry and Wildlife Management, Inland Norway University of Applied Sciences, N-2480 Koppang, Norway

**Keywords:** hepatitis E virus, *Hepeviridae*, Rangifer, serology, zoonosis

## Abstract

Hepatitis E virus (HEV) is a common cause of viral hepatitis in humans. In developing countries, HEV-infections seem to be mainly associated with pigs, but other animal species may be involved in viral transmission. Recently, anti-HEV antibodies were detected in Norwegian wild reindeer. Here, we investigated anti-HEV seroprevalence in Norwegian semi-domesticated reindeer, animals in closer contact with humans than their wild counterparts. Blood samples (n = 516) were obtained from eight reindeer herds during the period 2013–2017 and analysed with a commercial enzyme-linked immunosorbent assay designed for detecting anti-HEV antibodies in livestock. Antibodies were found in all herds and for all sampling seasons. The overall seroprevalence was 15.7% (81/516), with adults showing a slightly higher seroprevalence (18.0%, 46/256) than calves (13.5%, 35/260, *p* = 0.11). The seroprevalence was not influenced by gender or latitude, and there was no temporal trend (*p* > 0.15). A positive association between the presence of anti-HEV antibodies and antibodies against alphaherpesvirus and pestivirus, detected in a previous screening, was found (*p* < 0.05). We conclude that Norwegian semi-domesticated reindeer are exposed to HEV or an antigenically similar virus. Whether the virus is affecting reindeer health or infects humans and poses a threat for human health remains unknown and warrants further investigations.

## 1. Introduction

Hepatitis E virus (HEV) is a single-stranded positive sense RNA virus that belongs to the virus family *Hepeviridae*. While the icosahedral virions shed in the faeces are non-enveloped, the virions circulating in the blood are cloaked in host cell membranes and are considered to be “quasi-enveloped“ [1]. The HEV species Orthohepevirus A is divided into eight genotypes (GT), and HEV genomes of GT1-4 and GT7 have been detected in humans [2,3]. HEV GT1 and GT2 seem to be restricted to humans and cause sporadic hepatitis and outbreaks in Africa and Asia due to faecal contamination of drinking water [4]. HEV GT3 and GT4 are mainly found in domesticated pigs and wild boars, but have also been detected in camels, deer, rabbits, mongooses, cattle, goats and yaks [2,3]. In industrialized countries, at least in Europe and America, sporadic human infections by GT3 and, to a lesser extent, GT4 are believed to occur by ingestion of undercooked meat or milk, or through close contact with infected animals [5]. HEV GT7 has been found in camels in the Middle East and is shown to infect humans that consume camel milk and meat [6]. Transmission of HEV can also occur via consumption of contaminated berries, vegetables or shellfish [7,8,9], or via blood transfusions [10] or organ transplantations [11], or by vertical transmission from mother to foetus [12]. Recently, some human cases with HEV of species Orthohepevirus C, which usually infects rats, has been reported [13,14,15].

In most healthy individuals, the acute HEV-infection is either asymptomatic or takes a self-limiting course. For those that develop a systemic infection, a short prodromal phase with unspecific symptoms is followed by itching and liver specific symptoms such as jaundice, uncoloured stools and darkened urine [1,16]. The consequence is influenced by the genotype. Infection with HEV GT1 or GT2 may be aggravated for pregnant women as this increases the risk of preterm delivery, low birth weight, stillbirth and maternal death [4,12]. Infection with HEV GT3 or GT4 may lead to acute-on-chronic liver failure (ACLF) in patients with pre-existing liver disease, and to chronic liver infection in immunocompromised individuals such as transplant recipients, individuals infected by human immunodeficiency virus (HIV) and patients with hematologic malignancy undergoing chemotherapy [17]. Importantly, after only a few years of chronic HEV infection, progressive liver fibrosis and cirrhosis is commonly found [18]. In addition to this, HEV may cause a variety of extrahepatic manifestations such as neurologic diseases, acute pancreatitis, membranous glomerulonephritis, acute thyroiditis or hematologic abnormalities [19,20].

In Europe, anti-HEV seroprevalence rates between 0.035% and 60.9% have been reported in humans [21,22]. Two human seroprevalence studies have been performed in Norway. In the first study, 1200 blood donors and 79 swine farm workers were tested for anti-HEV antibodies, and a seroprevalence of 13.5% and 30.4%, respectively, was found [23]. In a population-based study of 1800 adults in Northern Norway, a seroprevalence of 11.4% was detected [24]. Importantly, a European meta-analysis found the highest seroprevalence in individuals exposed to pigs and/or wild animals [25]. As pig farming is not so common in Norway, especially not in the northernmost part, other animals may be more important for transmission to humans. A seroprevalence study of wild ungulates in a neighbouring country, Sweden, detected anti-HEV antibodies in 27.5% (19/69) of wild moose (*Alces alces*), 15.1% (21/139) of wild boar (*Sus scrofa*) and in a few roe deer (2/30, *Capreolus capreolus*) and red deer (1/15, *Cervus elaphus*) [26]. A similar study in Norway detected anti-HEV antibodies in 19.5% (32/164) of moose and 23.1% (43/186) of wild reindeer (*Rangifer t. tarandus*) [27], while a study on semi-domesticated reindeer (*Rangifer tarandus*) in Russia found anti-HEV IgG in 12.1% (23/191) of examined animals [28].

In Norway, there are approximately 25,000 wild reindeer and 220,000 semi-domesticated reindeer [29]. Reindeer herding is mainly conducted by indigenous Sami herders and is of major importance for their livelihood and culture [30]. Although reindeer husbandry is mostly based on natural pastures, the practice of supplementary feeding is increasing, which may increase nose-to-nose contact between the animals and contact between reindeer and humans. While reindeer meat used to be a traditional food mainly used by Sami people, it is now commonly consumed by many people and could potentially be a source for zoonotic HEV transmission in Norway. In the present study, we investigate, for the first time, anti-HEV seroprevalence in semi-domesticated Norwegian reindeer. The animals examined represent the geographically different reindeer herding regions in Norway. The results showed that all reindeer herds had been exposed to HEV and demonstrated an overall yearly seroprevalence fluctuating between 12.9% and 17.5%.

## 2. Results

### 2.1. Anti-HEV Seroprevalence in Semi-domesticated Reindeer

In order to analyse serum samples from Norwegian semi-domesticated reindeer, a commercial HEV ELISA intended for the detection of antibodies to HEV in serum of animals was used. An overall seroprevalence of 15.7% (81/516) was found (Table 1). The seroprevalence was slightly higher in adults than in calves, with antibodies detected in 18.0% (46/256) of the adults and 13.5% (35/260) of the calves, but this difference was not statistically significant (*p* = 0.11) (Table 1). Similarly, there was no significant difference between genders (*p* = 0.38), with anti-HEV antibodies detected in 15.7% (53/337) of females and 15.6% (28/179) of males. Anti-HEV antibodies were found in samples from all sampling seasons (Table 2) and from all eight herds (Table 1). The seroprevalence in the different herds varied between 6.7% and 33.8% (Table 1), with a median seroprevalence of 12.5%. There was no clear correlation between seroprevalence and latitude, as the highest seroprevalence was detected in animals from the Tromsø region, and the lowest seroprevalence was found in animals in the nearby Lødingen region (Table 1, Figure 1 and Figure 2A). The anti-HEV seroprevalence for each of the sampling seasons varied from 16.9% in 2013 to 17.5% in 2017, with no clear temporal trend (Table 2, Figure 2B). Of note, the number of samples included per year varied from 177 samples in 2013 to only 40 samples in 2017 (Table 2). We conclude that semi-domesticated reindeer herds representing the different reindeer herding regions of Norway are exposed to HEV.

### 2.2. Seroprevalence of Previously Examined Viruses

As some of the serum samples had recently been found to contain antibodies against pestivirus, alphaherpesvirus (i.e., Cervid herpesvirus 2; CvHV2) and/or gammaherpes-virus (viruses in the malignant catarrhal fever group; MCFV) [31], we investigated if there was an association between the HEV serostatus of a sample and the presence of antibodies against the previously investigated viruses. Overall, only small differences of 1.2–1.5-fold were detected between HEV-seropositive and HEV-seronegative animals for the three viruses (Table 3). A multiple correspondence analysis showed that the serostatus for pestivirus and HEV was correlated (chi-square test, *p* = 0.01), as well as for HEV and CvHV2 (chi-squared test, *p* = 0.05). The presence of gammeherpesvirus-antibodies, on the other hand, was not significantly associated with the presence of HEV-antibodies. We conclude that there was a correlation between being seropositive for HEV and pestivirus, and also between being seropositive for HEV and CvHV2.

## 3. Discussion

This is the first investigation of anti-HEV seroprevalence in semi-domesticated reindeer in Fennoscandia. The study revealed that infections by HEV, or an antigenically similar virus, are common in semi-domesticated reindeer living from latitude 60.91° N to 70.2° N in Norway. Anti-HEV antibodies were found in samples from all sampling years and in all herds examined. The total seroprevalence was 15.7%, with only minor yearly variation. This is slightly lower than the 23.1% reported for Norwegian wild reindeer, the latter mainly living around latitude 58.4° N to 62.4° N and sampled during the period 2010–2018 [27]. In contrast, the seroprevalence in semi-domesticated reindeer was slightly higher than the 12.1% reported for semi-domesticated reindeer sampled during the period 2018–2019 in eastern Russia [28]. While the Norwegian study of wild reindeer was based on the same ELISA as used in the present study, an ELISA detecting anti-HEV IgM, IgA and IgG, the Russian study used an ELISA only detecting anti-HEV IgG, which may indicate that these screenings are not necessarily directly comparable. Finally, comparing our results with the results of a recently published study on the same samples, addressing other viruses [31], the serostatus for HEV and pestivirus and HEV and CVHV2 were found to correlate.

The seroprevalence in adults was only slightly higher than in calves, with 18.0% versus 13.5%, respectively. This pattern has also been revealed in humans, where anti-HEV seroprevalence increase with age [24,25]. The results presented herein suggest that the majority of reindeer are infected relatively early in life, similar to what is reported for HEV in pigs. Pigs become infected from about seven weeks of age, when maternal antibody titres are declining, and HEV excretion is highest when they are 3–5 months of age [32]. Our results further suggest that the anti-HEV antibodies in reindeer are rather long-lasting, as we still detect antibodies in adult animals. Alternatively, the antibodies wane over time, but some animals undergo reinfection, as previously described for humans [33]. Another possible explanation for the relatively small difference in seroprevalence among calves and adult reindeer could be that many of the reindeer defined as adults (i.e., >1 year), were quite young. At sampling in October–April, the adults were 17 months or older. Unfortunately, we do not know the exact age as this was not determined. In humans, the persistence and protective role of anti-HEV antibodies is still unclarified. Anti-HEV IgG levels seem to steadily decline, but have in some cases been reported to persist up to 12 years after infection [34,35].

In contrast to humans, ungulates do not actively transfer immunoglobulins from the mother to the developing foetus. Instead, the new-borns acquire maternal immuno-globulins, including IgG, IgM, IgA and IgE, during the first colostrum feeding, and this is freely diffusing from the gastrointestinal tract to the blood stream during the first 1–2 days following birth [36]. Because the calves were already between five to 10 months old at sampling, we can exclude that the measurable amounts of anti-HEV antibodies at sampling was a result from maternal antibody transmission through colostrum.

The seroprevalence of the different herds varied from 6.7% to 33.8%, with a median value of 12.5%. It is not clear why the seroprevalence in Tromsø was much higher than in the other herds, especially because Tromsø had the lowest proportion of adults, only 17.6% (all females), versus 82.4% calves. For the other herds, the proportion of adults varied from 35.4% to 69.4%. A high population density and the gathering of wild ungulates at feeding spots have been suggested as general risk factors for infectious diseases [37]. All reindeer in our study were free ranging, but all herds except the two southernmost herds (Røros and Filefjell) obtained some supplementary feeding, thereby potentially increasing the risk of virus transmission.

In five of the herds (Tana, Tromsø, Lødingen, Hattfjelldal and Røros), the serostatus of the animals for pestivirus and CvHV2 seemed to correlate with the serostatus for HEV. The mechanisms for co-infection or potential facilitation are unclear. Detection of antibodies against different viruses in the same serum sample may be due to co-infections or infections separated in time. Co-infections of HEV and Porcine reproductive and respiratory syndrome virus (PRRSV) and HEV and Porcine circovirus type 2 (PCV2) in pigs have been reported [38]. As both PRRSV and PCV2 are immunomodulating pathogens, the co-infections were associated with a longer period of HEV shedding. If the reindeer had co-infections with HEV and pestivirus or HEV and CvHV2, this may have influenced HEV infection, transmission and possibly pathogenesis. So far, we do not know if reindeer are clinically affected by HEV-infection. Of note, post-mortem examination of experimentally HEV-infected pigs without evident clinical disease, detected multifocal lymphoplasmacytic hepatitis and focal necrotic hepatocytes consistent with mild viral hepatitis, demonstrating HEV-induced pathology [39].

Because the seroprevalence of HEV in reindeer was quite stable over the sampling seasons investigated (2013–2017), HEV seems to be enzootic in reindeer. Originally transmission of HEV to semi-domesticated reindeer may have occurred by indirect or direct contact with other infected animals. Because HEV is non-enveloped outside the body, it is stable and will stay infectious in water and soil for a long time after shedding. Wild boar have been suggested as the most important wildlife animal reservoir [40,41]. Only one to two decades ago there were no wild boar in Norway, but animals have crossed the border from Sweden in the south and today 400–1200 animals are estimated to be living in the south-eastern part of Norway. Some of them are infected by HEV, as shown by the detection of anti-HEV (1/86) in a recent study [42]. Although stray wild boars occasionally may be in the same area as herd 7 (Røros), they have never been observed in the areas close to the other herds and therefore probably played no important role in the transmission of HEV to reindeer. Several studies from continental Europe show that HEV infect rabbits, hares and rats [43,44]. Hares and rats could potentially have been involved in the transmission of HEV to semi-domesticated reindeer.

We cannot exclude the possibility that humans may have infected the semi-domesticated reindeer; however, this is less likely. From studies in pigs, we know that the shedding of HEV in faeces typically lasts 9.7–23.3 days [38], giving infected animals good opportunities to spread the virus to other animals. At this point, we cannot explain why only a relatively small number of animals in each herd appear to become infected.

An important question is whether HEV infecting semi-domesticated reindeer can infect humans, similar to what has been reported for HEV transmission from Sika deer (*Cervus nippon*) to humans [5]. In order to answer this, it is crucial to isolate HEV from reindeer, sequence the HEV genomes and compare this to HEV genomes detected in humans. In the study of semi-domesticated reindeer in Russia, there were unsuccessful attempts to detect HEV RNA in serum, and no other samples were investigated [28]. However, the authors noted that the seroprevalence in reindeer herders was similar to the seroprevalence of other adults in the area, suggesting that occupational infection did not play a major role. With the exception of reindeer herders, most people are seldom in direct contact with reindeer. In Norway, approximately 1100 tons of reindeer meat is consumed each year. Transmission could possibly occur via consumption of undercooked or raw meat (e.g., roasted, dried, or prepared as carpaccio) or dried/non-heated reindeer liver, the latter being marketed as a food supplement. An internal temperature of 71 °C for 20 minutes is apparently needed to inactivate infectious HEV [45]. In addition to the consumption of meat or liver, handling of the carcass may potentially lead to zoonotic infections, as butchers and slaughterhouse workers are reported to have an increased anti-HEV seroprevalence compared to other people in the same area (reviewed in [46]). In addition, faecal contamination of berries, mushrooms or drinking water could transmit the virus to humans.

In most identified human cases of HEV-infection, the source of infection remains unknown. Our finding of anti-HEV antibodies in serum from 15.7% of semi-domesticated reindeer suggest that HEV, or an antigenically similar virus, have infected these animals. So far, we have no indication that HEV is pathogenic for these animals. In order to find out if this virus may infect humans, a molecular characterization of HEV RNA from infected semi-domestic reindeer is urgently needed. Furthermore, a HEV-seroprevalence analysis of reindeer herders is required.

## 4. Materials and Methods

### 4.1. Animals and Sampling

Eight different semi-domesticated reindeer herds were sampled during late fall and winter (October–April) in two to five consecutive winter seasons, during the period 2013–2017 (i.e., the winters of 2013–2014, 2014–2015, and so forth). The herds were distributed in Norway from latitude 60.91° N to 70.2° N, with five of the herds (Tana, Lakselv, Tromsø, Lødingen and Hattfjelldal) in Northern Norway, two of the herds (Fosen and Røros) in Central Norway and one herd (Filefjell) in Eastern Norway (Figure 1). Whenever possible, serum samples were obtained from both calves (5–10 months old; both sexes) and adult females (>17 months old) from each of the eight herds, whereas adult bulls were sampled to a lesser extent due to restricted availability. A total of 516 serum samples from 256 adults (49.6%) (220 females and 36 males) and 260 calves (50.4%) (124 female and 136 males) were included (Table 4). The samples were mainly obtained from live animals during routinely herding practices, where the animals were released back into the herd after sampling (444/516; 86.0%), but also from dead animals during slaughter (72/516; 14.0%) (Table 5). Blood samples were collected from the jugular vein in blood tubes (BD Vacutainer®; BD, Plymouth, UK), using a venoject needle (Terumo, Leuven, Belgium) for live animals, or by collecting blood directly into open tubes during bleeding of slaughtered animals. Serum was prepared by centrifugation (10 min, 3000 × g) and stored at –20 °C until further analyses. The serum had been thawed a few times prior to the HEV antibody analysis for the previous seroprevalence study of alphaherpesvirus, gammaherpesvirus, pestivirus, bluetongue virus and Schmallenberg virus [31].

### 4.2. Serology

Serum samples (n = 516) were analysed for anti-HEV antibodies using a commercial double antigen sandwich enzyme-linked immunosorbent assay (ELISA) (HEV ELISA 4.Ov, MP Diagnostics, Eschwege, Germany), intended for the detection of antibodies to HEV in serum or plasma from swine and other animals such as wild boar, red deer and reindeer [27,47,48]. This ELISA uses a proprietary recombinant antigen, which is highly conserved between different HEV genotypes, to detect specific HEV-antibodies, including IgG, IgM and IgA. Specimens with absorbance values greater than or equal to the cut-off value (0,20 absorbance unit plus the mean absorbance of a negative control) were considered initially reactive but were retested in duplicate before interpretation. Only samples found to be reactive on retesting, were interpreted to contain anti-HEV antibodies.

### 4.3. Statistical Analyses

Seroprevalence of anti-HEV antibodies were analysed using generalized linear models in R 4.1.0 [49] with a binomial outcome (negative/positive result in the test, coded as 0 or 1, respectively) and a logit link. Each individual’s age group (calf or adult), latitude, sex and sampling season were used as explanatory variables; the significance level was set to *p* < 0.05 for the explanatory variables. In order to discover a relation between the HEV serostatus and the serostatus of the previously investigated viruses [31], we performed a multiple correspondence analysis [50] using year, latitude, sex, age and the serostatus for these viruses as contrasts. Apparent correlations were further studied with Chi-squared tests to conclude if there was a correlation between the different infections.

## Figures and Tables

**Figure 1 pathogens-10-01542-f001:**
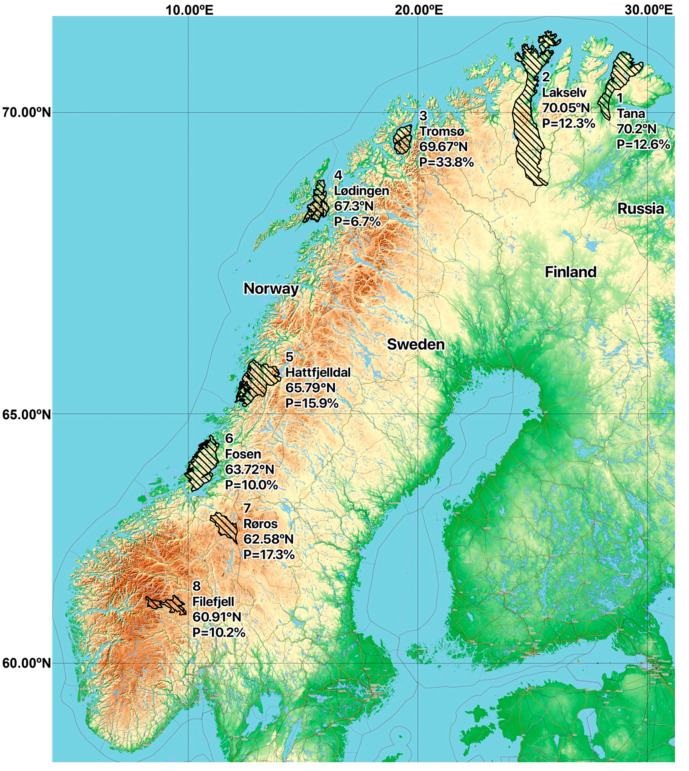
Map showing the geographical pasture positions of the eight sampled Norwegian herds of semi-domesticated Eurasian tundra reindeer (*Rangifer tarandus tarandus*) that were screened for the presence of anti-HEV antibodies (n = 516; 2013–2017). The anti-HEV seroprevalence found for each herd is shown. Map created using the Free and Open Source QGIS. Map data: ©OpenStreetMap-Mitwirkende, SRTM | Map position: ©OpenTopoMap (CC-BY-SA). Reindeer pastures data source: NIBIO.

**Figure 2 pathogens-10-01542-f002:**
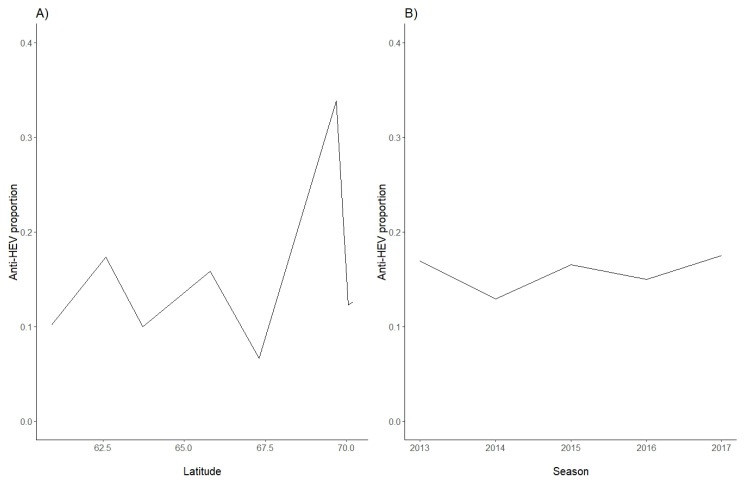
Seroprevalence proportions in Norwegian semi-domesticated reindeer in herds, sorted by (**A**) latitude and (**B**) sampling season (2013–2017).

**Table 1 pathogens-10-01542-t001:** Anti-HEV seropositive animals in different herds of semi-domesticated reindeer.

Location of Herd	Adults		Calves		Seropositive Animals (%)
	Females	Males	Females	Males	
1. Tana	10/65	0/1	1/19	1/10	12/95; 12.6%
2. Lakselv	5/33	1/5	1/11	0/8	7/57; 12.3%
3. Tromsø	5/12	0/0	5/23	13/33	23/68; 33.8%
4. Lødingen	1/15	0/2	0/2	1/11	2/30; 6.7%
5. Hattfjelldal	6/26	2/3	2/19	3/34	13/82; 15.9%
6. Fosen	4/24	0/6	1/4	0/16	5/50; 10.0%
7. Røros	7/31	1/5	3/21	2/18	13/75; 17.3%
8. Filefjell	1/14	3/14	1/16	1/15	6/59; 10.2%
Total	39/220; 17.7%	7/36; 19.4%	14/117; 12.0%	21/143; 14.7%	81/516; 15.7%
	46/256; 18.0%		35/260; 13.5%		

**Table 2 pathogens-10-01542-t002:** Seroprevalence of HEV in semi-domesticated reindeer per year.

Year of Sampling	Number of Animals Included	Seropositive Animals (%)
2013	177	30 (16.9%)
2014	124	16 (12.9%)
2015	115	19 (16.5%)
2016	60	9 (15.0%)
2017	40	7 (17.5%)
Total	516	81 (15.7%)

**Table 3 pathogens-10-01542-t003:** Seroprevalence of other viruses in HEV-seronegative and HEV-seropositive semi-domesticated reindeer.

	Herd	In HEV Seronegative Animals	In HEV Seropositive Animals	Fold in HEV Seropositive/ Seronegative Animals
Pestivirus seropositive n = 89 (5 herds)	Tana n = 95	12/83 (14.5%)	2/12 (16.7%)	1.2
	Tromsø n = 68	2/45 (4.4%)	4/23 (17.4%)	4.0
	Lødingen n = 30	3/28 (10.7%)	0/2	-
	Hattfjelldal n = 82	26/69 (37.7%)	11/13 (84.6%)	2.2
	Røros n = 75	24/62 (38.7%)	5/13 (38.5%)	1.0
	TOTAL n = 350	67/287 (23.3%)	22/63 (35.0%)	1.5
Alpha-herpesvirus seropositive n = 221 (All 8 herds)	Tana n = 95	49/83 (59%)	10/12 (83.3%)	1.4
	Lakselv n = 57	26/50 (52%)	4/7 (57.1%)	1.1
	Tromsø n = 68	7/45 (15.6%)	7/23 (30.4%)	1.9
	Lødingen n = 30	11/28 (39.3%)	0/2	-
	Hattfjelldal n = 82	22/69 (31.9%)	7/13 (53.8%)	1.7
	Fosen n = 50	21/45 (46.7%)	2/5 (40%)	0.9
	Røros n = 75	21/62 (33.9%)	8/13 (61.5%)	1.8
	Filefjell n = 59	21/53 (35.6%)	5/6 (83.3%)	2.3
	TOTAL n = 516	178/435 (40.9%)	43/81 (53.0%)	1.3
Gamma-herpesvirus seropositive n = 54 (All 8 herds)	Tana n = 95	3/83 (3.6%)	1/12 (8.3%)	2.3
	Lakselv n = 57	7/50 (14.0%)	1/7 (14.3%)	1.0
	Tromsø n = 68	3/45 (6.7%)	1/23 (4.3%)	0.6
	Lødingen n = 30	1/28 (3.6%)	0/2	-
	Hattfjelldal n = 82	10/69 (14.5%)	3/13 (23,1%)	1.6
	Fosen n = 50	7/45 (15.6%)	0/5	-
	Røros n = 75	6/62 (9.7%)	4/13 (30.8%)	3.1
	Filefjell n = 59	7/59 (11.9%)	0/6	-
	TOTAL n = 516	44/435 (10.1%)	10/81 (12.3%)	1.2

**Table 4 pathogens-10-01542-t004:** Semi-domesticated reindeer included in the study.

Location	Year of Sampling	Adults			Calves			Animals per Year
		Females	Males	Total	Females	Males	Total	
1. Tana	2013	19	1	20	0	0	0	20
	2014	16	0	16	1	0	1	17
	2015	11	0	11	4	3	7	18
	2016	10	0	10	4	6	10	20
	2017	9	0	9	10	1	11	20
	Total	65	1	66 (69%)	19	10	29 (21%)	95
2. Lakselv	2013	9	3	12	4	2	6	18
	2014	14	0	14	4	1	5	19
	2015	10	2	12	3	5	8	20
	Total	33	5	38 (67%)	11	8	19 (23%)	57
3. Tromsø	2013	0	0	0	7	18	25	25
	2014	5	0	5	10	8	18	23
	2015	7	0	7	6	7	13	20
	Total	12	0	12 (18%)	23	33	56 (82%)	68
4. Lødingen	2013	7	2	9	11	2	13	22
	2014	8	0	8	0	0	0	8
	Total	15	2	17 (57%)	11	2	13 (43%)	30
5. Hattfjelldal	2013	6	0	6	7	12	19	25
	2015	2	1	3	0	14	14	17
	2016	8	2	10	5	5	10	20
	2017	10	0	10	7	3	10	20
	Total	26	3	29 (35%)	19	34	53 (65%)	82
6. Fosen	2013	6	3	9	1	10	11	20
	2014	12	2	14	3	3	6	20
	2015	6	1	7	0	3	3	10
	Total	24	6	30 (60%)	4	16	20 (40%)	50
7. Røros	2013	3	2	5	8	9	17	22
	2014	15	0	15	4	0	4	19
	2015	7	0	7	5	2	7	14
	2016	9	1	10	3	7	10	20
	Total	31	5	36 (48%)	21	18	39 (52%)	75
8. Filefjell	2013	5	7	12	7	6	13	25
	2014	7	3	10	4	4	8	18
	2015	2	4	6	5	5	10	16
	Total	14	14	28 (47%)	16	15	31 (53%)	59
	Total	220	36	256	124	136	260	516

**Table 5 pathogens-10-01542-t005:** Sampling of semi-domesticated reindeer.

	Live	Slaughtered
Calves (n = 260)	219 (84.2%)	41 (15.8%)
Adults (n = 256)	225 (87.9%)	31 (12.1%)
Total (n = 516)	444 (86.0%)	72 (14.0%)

## Data Availability

The datasets presented in this article are not readily available because the data on which the article is based on contains personal data on identifiable reindeer herders and their animals. Requests to access the dataset should be directed to the corresponding authors.

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
