# Peer review of "Serological Evidence of Hepatitis E Virus Infection in Semi-Domesticated Eurasian Tundra Reindeer (Rangifer tarandus tarandus) in Norway"

_pathogens, 2021, doi:10.3390/pathogens10121542_

Round 1

Reviewer 1 Report

Dr. Rinaldo and colleagues presented an interesting retrospective observational study in which they aimed to investigate the seroprevalence of anti-HEV antibodies in semi-domesticated Eurasian tundra reindeer in Norway. The colleagues collected blood samples from reindeer of different herds between 2013 and 2017 and analysed these using a commercial ELISA (IgG, IgM, and IgA ELISA kit by MP). The authors could determine an overall seroprevalence of 15.7% (6.7%-33.8%) in the eight reindeer herds with no statistically significant differences between gender, age, and latitude.

Overall the manuscript is well written and concise in its content. The experiments are well designed and the results are convincing. The supplementary information adequately supports the main text. The English is adequate and fine and the references are up-to-date. The figures and tables are well designed and understandable.

Comments

  1. The introduction section is quite long and should be trimmed. Please, focus on the key aspects of the article.
  2. The main question that arises from this study is the HEV genotype. Were there animals with acute infection (IgM positive) from which viral RNA could possibly be isolated and further analyzed?
  3. The relationship between co-infection / detection of antibodies of HEV with herpesvirus and pestivirus in the reindeer is not clear to me. What is the significance of this?

Reviewer 2 Report

In this paper, the authors have investigated the presence of anti-HEV antibodies in semi-domesticated reindeer in Norway and found an overall seroprevalence of 15%.

Overall, this research article is of particular interest for the field as it describes the seroprevalence in semi-domesticated reindeer in Norway. This study gives valuable data on the potential role of reindeer as HEV reservoir even if more investigations are required to characterize the strains circulating in these animals.

Minor comments:

  • Line 40-41: More cases of Hepatitis E virus species C infection in humans have been reported recently. The reference cited (2) is also not appropriate.
  • Figure 1: Police size could be increased for clarity.
  • Figure 2 : It is not easy to read this figure (police size too small). I wonder if this figure is useful. Data presented in the right panel are already clearly presented in table 2. Data presented in the left panel are included in table 1 and figure 1. It could be more useful to remove figure 2 and add seroprevalence data (%) into figure 1.
  • Table 3: Pestvirus (and not pestisvirus). Are the sample size of the different groups sufficient to perform statistical analysis?
  • Figure S1: Figure is not clear. Please increase police size. I found this figure difficult to follow. Is it not possible to remove this supplementary figure and instead include the results of the statistical analysis in table 3 ?
  • It could be useful to provide a brief description of the assays used to determine the seroprevalence of pestivirus, alphaherpesvirus and gammaherpesvirus and antibodies against which viruses they detect.
  • Line 173: Please add the seroprevalence found for the other types of virus. It would be easier to compare the data.
  • Table 5 displayed before table 4.
  • Line 178-179: Re-infection might also occur in adult animals?
  • Line 205-206: What do you mean by “obtained various degree of feeding”?
  • Line 230: “Stray wild boars” rather than “stray animals of wild boars”
  • Line 242 : “ An important question is whether HEV ..”
  • Line 256: Market instead of “marked”
  • Line 272: Required rather than “desired”
